# Association of Progranulin Gene Expression from Dyspeptic Patients with Virulent *Helicobacter pylori* Strains; In Vivo Model

**DOI:** 10.3390/microorganisms10050998

**Published:** 2022-05-10

**Authors:** Claudia Troncoso, Mónica Pavez, Álvaro Cerda, Victor Manríquez, Aurora Prado, Edmundo Hofmann, Eddy Ríos, Armando Sierralta, Luis Copelli, Leticia Barrientos

**Affiliations:** 1Doctoral Program in Sciences, Mention in Applied Cellular and Molecular Biology, Universidad de La Frontera, Temuco 4811230, Chile; troncosomunozc@gmail.com; 2Faculty of Health Sciences, Universidad Autónoma de Chile, Temuco 4810101, Chile; 3Laboratory of Clinical Microbiology Research, Center of Excellence in Translational Medicine, Universidad de La Frontera, Temuco 4801019, Chile; 4Laboratory of Bioanalysis and Molecular Diagnosis, Center of Excellence in Translational Medicine, Universidad de La Frontera, Temuco 4801019, Chile; alvaro.cerda@ufrontera.cl (Á.C.); v.manriquez02@ufromail.cl (V.M.); 5Laboratory of Applied Molecular Biology, Center of Excellence in Translational Medicine, Scientific and Technological Nucleus in Bioresources (BIOREN), Universidad de La Frontera, Temuco 4801019, Chile; a.prado01@ufromail.cl; 6Department of Internal Medicine, Universidad de La Frontera, Temuco 4781218, Chile; edmundo.hofmann@ufrontera.cl (E.H.); eddy.rios@ufrontera.cl (E.R.); armando.sierralta@gmail.com (A.S.); 7Gastroenterology Unit of Clínica Alemana de Temuco, Temuco 4810297, Chile; 8Gastroenterology Unit of Hospital Hernán Henríquez Aravena, Temuco 4781151, Chile; 9Gastroenterology Unit of Hospital de Villarrica, Villarrica 4930000, Chile; luis.coppelli@gmail.com

**Keywords:** progranulin, *Helicobacter pylori*, virulence, gastric lesions

## Abstract

(1) Background: Gastric cancer, the fourth most common cause of death from tumors in the world, is closely associated with *Helicobacter pylori*. Timely diagnosis, therefore, is essential to achieve a higher survival rate. In Chile, deaths from gastric cancer are high, mainly due to late diagnosis. Progranulin has reflected the evolution of some cancers, but has been poorly studied in gastric lesions. Aiming to understand the role of progranulin in *H. pylori* infection and its evolution in development of gastric lesions, we evaluated the genic expression of progranulin in gastric tissue from infected and non-infected patients, comparing it according to the epithelial status and virulence of *H. pylori* strains. (2) Methods: The genic expression of progranulin by q-PCR was quantified in gastric biopsies from Chilean dyspeptic patients (*n* = 75) and individuals who were uninfected (*n* = 75) by *H. pylori*, after receiving prior informed consent. Bacteria were grown on a medium Columbia agar with equine-blood 7%, antibiotics (Dent 2%, Oxoid^TM^), in a microaerophilic environment, and genetically characterized for the *ureC*, *vacA*, *cagA*, and *iceA* genes by PCR. The status of the tissue was determined by endoscopic observation. (3) Results: Minor progranulin expression was detected in atrophic tissue, with a sharp drop in the tissue colonized by *H. pylori* that carried greater virulence, VacAs1m1^+^CagA^+^IceA1^+^. (4) Conclusions: Progranulin shows a differential behavior according to the lesions and virulence of *H. pylori*, affecting the response of progranulin against gastric inflammation.

## 1. Introduction

*H. pylori* is the principal bacteria associated with several gastric pathologies. Over 50% of the world’s population is colonized by *H. pylori* [1,2,3]. *H. pylori* can colonize gastric epithelial cells through various mechanisms, causing several gastro-duodenal diseases, including gastritis, ulcers, gastric cancer, and mucosa-associated lymphoid tissue (MALT) lymphoma [4]. These gastro-duodenal diseases are influenced by multiple factors that determine the severity of lesions, such as flagellar structure, adherence capacity and urease production, among other bacterial conditions. Additionally, genetic background, host response, and environmental factors can produce harmful effects on the epithelium infected by *H. pylori* [5]. Furthermore, the bacteria express a diversity of virulence factors, particularly VacA cytotoxin and CagA oncogenic protein, that are associated with higher severity of infection [6,7]. All *H. pylori* strains carrying the *vacA* gene are associated with cell vacuolation and modulation in mitochondrial and membrane disruption, resulting in Cytochrome C release, leading to apoptosis and alterations in cellular signaling pathways [8]. In addition, after entering the gastric epithelial, the cell Cag A protein is phosphorylated, producing intracellular perturbation, the activation of actin, stimulation of inflammatory responses, disruption of tight junctions, and a modification of the cellular proliferation [5,9]. Other virulence factors such as adhesins proteins favor bacterial colonization, causing inflammation and distress of the epithelium, as is the case with IceA, which is linked to gastric ulcers [10]. These changes in the mucosa induce host reactions such as immune and inflammatory responses, activating inflammatory mediators, macrophages, cytokines, and other elements of cellular response, including cell proliferation pathways and signaling pathways such as Mitogen-activated Protein Kinases (MAPK) involved with diverse physiology cell mechanisms such as motility, adhesion, metabolic, mitosis, apoptosis, and cellular differentiation [11,12,13]. As *H. pylori*-inducing cell proliferation leads to modifications, through the MAPK pathway by p38 and MEK1/2, and the progranulin protein (PGRN) responds to gastric tissue inflammation through similar mechanisms [14,15]. PGRN, also called granulin-epithelin precursor (GEP), proepithelin, GP88, PC-cell-derived growth factor (PCDGF), and acrogranin [16,17] is a glycosylated protein of 598 amino acids and 88 kDa (~65 kDa without glycosylation). The gene for PGRN is designated as *GRN* in humans and localized on chromosome 17q21.31 [18]. PGRN is a functional precursor containing a signal sequence and seven and one half granulin-like domains, connecting the conservative granulin (A, B, C, D, F, G) units through a linking region (P1–7) with an extensive range of physiological functions [19]. It is involved in the promotion of epithelial cell proliferation and wound healing and is important in regulating inflammation, at least in part by directly binding to tumor-necrosis-factor receptors (TNFR) and counteracting the TNF-mediated inflammatory signaling pathway [20]. PGRN is degraded to granulin residue by matrix metalloproteinase (MMP), ADAMTS-7, elastase, and proteinase, linked to inflammatory effects [16]. Currently, PGRN is considered a growth factor with high expression levels in hepatocellular, ovarian, bladder cancer, and glioblastoma, and is associated with a bad prognosis [17]. Their dual behavior stands out as an anti-inflammatory response with a high expression in aggressive cancer cell lines associated with the intact or degraded status of the molecule [21]. However, studies of the relationship between PGRN and *H. pylori* infections have been conducted, evaluating the expression of PRGN in infected and non-infected tissues only with gastritis status lesions, and have produced discordant results among immunohistochemical and genic expression methods [22]. The aim of this study was to evaluate the genic expression of *GRN* in dyspeptic patients with and without *H. pylori* infection, comparing the gastric epithelial status, the several virulence profiles of *H. pylori*, and *GRN* behavior from gastric biopsies of patients in the south of Chile.

## 2. Materials and Methods

### 2.1. Patients, Clinical Specimens, and Design of the Study

A total of 150 dyspeptics patients, 75 of whom were positive and 75 negative to urease, over 18 years of age, who were patients in the Endoscopic Units from three health centers of Región de La Araucanía, Southern Chile, were included in this study. All subjects signed informed consent forms prior to participating in this study, in compliance with the Declaration of Helsinki, and the study was approved by the Scientific Ethics Committee from Universidad de La Frontera, Chile (Protocol no. 028/18, AS1). In addition, the participants responded to a survey for sociodemographic information (age, sex, educational level, Mapuche ethnicity, household, place of residence, and *H. pylori* infection history), in addition to morbidity characteristics, metabolic and cardiovascular diseases, and family history of gastric cancer. Patients were excluded from this research if they had received any treatment with non-steroidal or anti-inflammatory drugs, antibiotics, or proton pump inhibitors three weeks before the sampling.

Through an endoscopy procedure, three antral gastric biopsies were obtained from patients, and simultaneously, esophagus, stomach, and duodenum abnormality (gastritis, ulceration, erosion, and others) were recorded.

To generate case and control groups with similar characteristics, and thus eliminate variability caused by factors external to the study, both infected and non-infected groups were selected by matching people based on similarity of age, sex, and ethnicity. The infected group included urease-positive patients (*n* = 75), with or without *H. pylori* eradication treatment, and bacteria isolation through microbiological culture, which was confirmed by the *ureC* gene genotype. The non-infected group included urease-negative patients (*n* = 75) without a history of *H. pylori* eradication treatment, and negative microbiological culture for *H. pylori*.

### 2.2. Recollection of Specimens

From the three biopsies available, one biopsy was used for the Rapid Urease test in the respective Health Centre using the Rapid Urease Test (RUT) CLOtest^TM^, (Halyard, Alpharetta, GA, USA). For the microbiology analyses, the second sample was stored in a sterile environment in an Eppendorf tube with 200 µL of Brucella Broth (BD-DIFCO^TM^, Berkshire, UK). The third sample was recollected in cryotube with 500 µL of RNA later (QUIAGEN^TM^, Hilder, Germany), and conserved at −80 °C prior to *GRN* expression analyses. Finally, samples were transported in containers at 4 °C to the Applied Biology Molecular Laboratory of the Centre of Translational Medicine at the Universidad de La Frontera.

### 2.3. Categorization of the Gastric Epithelial Status

According to the gastric or duodenal mucosa status, there were four categories of the epithelial gastric status that were assigned by endoscopic observation, which are as follows: (1) Non-lesion (NL) for tissues without inflammatory or injuries signals, (2) Erosive lesion (EL) for tissues with wounds or ulcered, (3) Non-erosive lesion (NEL) for the simple inflammation or non-ulcerated epithelial injuries, and (4) Atrophic lesion (AtL), in the case of the atrophic epithelium [21,23]. 

### 2.4. Extraction of Total RNA and Relative Expression of Progranulin by q-PCR 

To obtain the ARN from biopsies we employed the miRNA Isolation Kit (mirVANA^TM^, Thermo Fisher Scientific, Pleasanton, CA, USA). The protocol was adjusted to the manufacturer’s recommendations by considering the size of the gastric samples. First, the tissues were manually crushed with a mortar, and the reaction volumes were adjusted to the mass of the samples, with approximate mass reaching an average of 5.0 mg. Total RNA quantity and quality were evaluated using the absorbance index 260/280 in Nanoquant M200 PRO, TECAN (Tecan Trading AG, Männedorf, Switzerland). Total RNA was reverse-transcribed to cDNA with oligo (dT), Random Primers assay Invitrogen^TM^, and SuperScript II Invitrogen^TM^. Finally, 20 µL of the reaction mix was exposed at 25 °C for 10 min, 42 °C for 50 min, and 70 °C for 15 min in the Labnet Digital Thermocycler Equip, Multigene Optimax model (LabNet International. Inc., Darmstadt, Germany), to obtain cDNA. 

The real-time PCR reaction involved the TaqMan Assay (TaqMan Gene Expression Assay (FAM) Biosystems^TM^, Foster City, CA, USA). Progranulin mRNA expression was analyzed in a duplicate *GRN* assay (Hs00963703_g1-Applied Biosystems^TM^, Foster City, CA, USA), the Glyceraldehyde 3- phosphate dehydrogenase assay (GAPDH) expression (Hs00266705_g1-Applied Biosystems^TM^, Foster City, CA, USA) as a housekeeping gene, and Master Mix PCR Universal TaqMan2x (Applied Biosystems^TM^, Foster City, CA, USA). The initial denaturation step at 95 °C for 10 min was followed by 37 cycles with a denaturation duration of 15 s at 95 °C, annealing 30 s at 58 °C and elongation at 45 s at 75 °C using 7500 Fast Dx Real-Time PCR Instrument, (Applied Biosystems™, Waltham, MA, USA). The fluorescence intensity reflected the amount of actually formed PCR-product. The genic expression was calculated with the comparative threshold cycle method, 2^−ΔCt^.

### 2.5. Isolation and Confirmation of Helicobacter pylori Species

The gastric tissue for culture was processed within a maximum of two hours. Initially, the tissues were manually grounded with a sterile polypropylene baguette until detached from the mucus tissue. The macerated tissues (100 µL) were inoculated on Columbia agar (Oxoid^TM^ CMO 331, Basingstoke, Hampshire, UK) plates enriched with 7% (*v*/*v*) equine blood and supplemented with the antibiotics Trimethoprim (5 µg mL^−1^), Amphotericin B (5 µg mL^−1^), Cefsulodin (5 µg mL^−1^), and Vancomycin (10 µg mL^−1^)—(Supplement DENT 2%, Oxoid^TM^, Basingstoke, Hampshire, UK)—to favor *H. pylori* growth. Tissues were incubated in a microaerophilia atmosphere (5% O_2_, 10% CO_2_, 85% N_2_, and 90% humidity) using the CampyGen closed system (Oxoid^TM^, Basingstoke, Hampshire, UK) for three to seven days at 37 °C. Visible bacterial colonies were separated according to their phenotype, morphology characteristics, Gram stain, Catalase, and Oxidase test. All pure isolates were stored at −80 °C prior to molecular identification.

The Polymerase Chain Reaction (PCR) was applied for confirmation of the *H. pylori* species. First, the DNA of isolated bacteria with the suggestive *H. pylori* phenotype was extracted using a microbial DNA extraction kit (DNeasy UltraClean Microbial Kit^®^ Qiagen, Hilden, Germany) following the manufacturer’s recommendations. Subsequently, PCR amplification of the *ureC* gene and 16SrRNA conserved region [24,25], (Appendix A) was performed. The PCR amplification assays were made using a reaction mix prepared with 2.5 µL of the PCR Buffer 10X, 2.5 µL of dNTPs 200 µM, 0.125 µL Taq polymerase 1.25U (BioLabs, Inc. NEB, Ipswich, MA, USA), 1 µL of the specific forward and reverse primer of each gene at 2.5 µM mL^−1^, 16.875 µL of molecular grade water, and 1 µL of DNA. Finally, the 25 µL reaction mixture was subjected to one denaturation cycle at 94 °C for 5 min, followed by 35 additional cycles, including denaturation at 94 °C for 30 s, hybridization at 53 °C for 30 s, extension at 72 °C for 30 s, and the last extension cycle at 72 °C for 5 min in a digital thermocycler, Multigene Optimax model (LabNet International. Inc. Merck KGaA, Darmstadt, Germany). Subsequently, the amplified DNA was visualized in 1% agarose gel/TBE 0.5X Buffer (Sigma-Aldrich, St. Louis, MO, USA), stained with Red Gel Ladder (Bio Labs Inc. NEB, Ipswich, MA, USA) and DNA Ladder molecular weight marker of 50 pb, by electrophoresis methods, and observed under a UV transilluminator MyEcl Image (Thermo Fisher, Pleasanton, CA, USA). The *H. pylori* ATCC 26695 was used as a positive control of the PCR assay. 

A patient was considered *H. pylori* infected when the rapid urease test was positive, isolates spiral-shaped bacillus bacteria and negative Gram stain were present, the Catalase and Oxidase test was positive, and when the presence of *16SrRNA* and *ureC* was detected.

### 2.6. Detection of Virulence Gene vacA and s/m Alleles, iceA More iceA1, iceA2 Alleles, and cagA, of the Helicobacter pylori Species by PCR and PCR-RFLP

The characterization of the virulence of *H. pylori* strains was carried out through the amplification of the *vacA*, *cagA*, and *iceA* genes by the conventional PCR technique, including the *s* and *m* alleles of *vacA* and the *A1* and *A2* of iceA alleles, using specific primers according to the respective virulence genes, detailed in Appendix A. The reaction mixture and amplification conditions previously described in item 2.7, were adjusted to the specific melting temperature according to each primer used (Appendix A). Finally, the PCR products were visualized on agarose gel by electrophoresis. 

Additionally, *vacA s/m* alleles were confirmed by RFLP (Restriction Fragment Length Polymorphism) using DNA restriction enzyme, BstUI (Thermo Scientific™, Carlsbad, CA, USA). The reaction mixture used in this assay was made up of 10 ul of amplified segment of *vacA* (alleles *s* and *m*, respectively), 2 uL of Buffer NE 10x (Merck^TM^, Darmstadt, Germany), 0.4 µL of the BstUI enzyme (Thermo scientific^TM^. Carlsbad, CA, USA) and 7.6 µL of molecular-grade water (BioLabs, Inc. NEB, Ipswich, MA, USA) [25]. The reaction mix was subjected to 60 °C for 5–10 min in a thermocycler. The DNA fragments were separated by agarose gel electrophoresis 1.5% (Cleaver Scientific, Rugby, UK), in Buffer 0.5 X TBE, for 90 min, 90 V, and 300 A, and visualized in transilluminator. 

### 2.7. Statistical Analyses

The sociodemographic records according to the infected (case) and uninfected groups (control), as well as the virulence characteristics of the *H. pylori* strains and the levels of expression of PGRN present in the different status of gastric tissue were analyzed with the statistical GraphPad Prisma 9.0 program. The data were expressed as average ± SD, percentages and mean and range. The continuous variables were analyzed by *t*-test. The categorical variables according to the sociodemographic characteristics between the infected and non-infected groups, and for virulence comparisons and gastric tissue status, were analyzed using Chi square and Fisher’s exact test. In addition, the *GRN* expression levels were analyzed by the non-parametric Mann–Whitney test to compare two groups, and the Kruskal–Wallis test for multivariable analyses, according to the gastric tissue status of infected and uninfected patients, and *H. pylori* strain characteristics. Statistical significance was established at *p* < 0.05, applying a 95% confidence interval. 

## 3. Results

### 3.1. Sociodemographic Characterization and Clinical Data of Study Group

According to the total number of participants (*n* = 150/100%) and sample selection criteria, the studied population presented an average age of 48.18 ± 14.26 (Table 1). Women were predominant (*n* = 102, 68.00%), and 46 (30.67%) participants were identified as belonging to the Mapuche ethnic group (a local ethnic group from Chile). Both groups—infected (case) and non-infected (control)—showed equal numbers of rural residents (*n* = 16, 21.33%), and similar public health coverage FONASA (from Spanish—“FOndo NAcional de SAlud”) (*n* = 59, 78.67%, and *n* = 58, 77.33%, respectively), as well as an education level of < 12 years (*n* = 44, 58.67% and *n* = 42, 56.00%, respectively). Interestingly, the control group reported greater comorbidities, such as hypercholesterolemia (*n* = 24, 32.00%, *p* = 0.035). 

### 3.2. Gastric Epithelium Status 

According to the endoscopic report, normal-appearing epithelium was predominantly observed, denominated as epithelia non-lesion (*n* = 94/62.67%), of which 46 samples (61.33%) corresponded to tissues infected with *H. pylori* (Table 2). Concerning the altered status of the epithelium, 56 (37.33%) biopsies presented some level of lesions, 29 (38.67%) belonging to the case group (infected by *H. pylori*). Within this group, 11 (14.67%) presented non-erosive lesions, with a significant difference in the non-infected (*n* = 2, 2.67%), of *p* = 0.010. Of the 35 participants with erosive lesions (EL) (23.33%), only 14 (18.67%) had *H. pylori*. The remaining were non-colonized by the bacteria, with significant differences (*p* = 0.029). In the atrophic lesions, the distribution of lesions was similar in the infected and non-infected groups, LAt *n* = 6, 4.00%, (*n* = 3, 4.00% belonging to infected group) with non-significant differences. It is important to mention that in advanced lesions (AdL), specifically metaplastic lesions, participants both in the infected group (cases) and in the non-infected group (controls), presented only one gastric tissue sample. For this reason, they were not analyzed.

### 3.3. Progranulin and Host Response to H. pylori Colonization and Virsulence

The *GRN* expression levels observed in Figure 1 show similitude in both the infected and non-infected groups, with non-significant differences. 

When observing the levels of PGRN gene expression in the gastric tissue, according to the presence or absence of lesions, these values remain similar between the infected and non-infected groups (Figure 2).

Furthermore, non-significant differences of the *GRN* expression between infected and non-infected, as was the case between the non-lesion (NL) or lesion tissue (L) status of the cases (infected) or controls (non-infected) were observed. A significant reduction in *GRN* expression for individuals infected with *H. pylori*, according to the progress of gastric lesions and based on the precancerous cascade by Correa [26], was observed (Figure 3). In non-erosive lesions (NEL) the *GRN* expression began to decrease in comparison with an NL or EL status. Furthermore, the decrease in *GRN* expression was more significative as it progressed towards precancerous lesions, with PGRN reaching the lowest expression in tissues with atrophic lesions (LAt) (*p* < 0.05). 

The behavior of the PGRN was different in non-infected tissue by *H. pylori,* showing a significant difference between AtL and SL tissue (*p* = 0.049). However, we could observe a more acute decrease in the *GRN* expression of infected group (cases) than in the non-infected (controls) groups, as well as progress toward serious injuries in the Correa Cascade.

### 3.4. Genetic Identification of Virulence Factors of Helicobacter pylori

The virulence characteristics of colonizing *H. pylori* strains in the gastric tissue were studied with the virulence factors VacA, CagA and IceA in the 75 strains belonging to the infected group (cases) (Figure 4). All the isolates presented VacA with a predominance of alleles of the m2 (41/75, 54.67%) and s2 (40/75, 53.33%) strains, respectively. The *iceA*, present in 38 (50.67%) strains, was the second-most frequent. The allelic *A2* profile was the most frequent (*n* = 20/26.67%), and four (*n* = 5.33%) strains presented both A1/A2 alleles. When observing the distribution of the *vacA* and *iceA* alleles, the most predominant were s2m2 (*n* = 29, 38.67%), which was distributed mainly in the tissue without lesions (*n* = 19, 41.30%). For *iceA*, the *A2* allele was the most predominant, as it was detected in 20 (52.63%) isolates, and 12 (*n* = 63.15%) colonizing strains in tissues with lesions. Regarding the *vacA* s1m1 alleles, which were more closely linked to gastric lesions, 11 (37.93%) strains were observed in tissues with lesions that were mainly erosive (LE, *n* = 5, 35.72%). The *A1* alleles were characterized by an association with superficial gastric lesions, with six (31.58%) isolates taken from tissues with lesions, 33.3% (*n* = 1) of in which colonized the AtL. The *cagA* was present in 45.33% of isolates with distribution in all epithelial conditions (Appendix A). 

### 3.5. Expression Levels of GRN mRNA in the Gastric Tissue, According to the Virulence Characteristics of H. pylori Strains

The gene expression of PGRN showed no changes according to the virulence of *H. pylori* (Figure 5).

When we evaluated the gene expression of PGRN by the presence or absence of CagA, the levels of *GRN* expression decreased in the tissues colonized with the CagA^+^ strains, although with significant non-differences (Figure 6).

The expression of *GRN* in the epithelium infected with *H. pylori* strains carrying *cagA*+ tends to decrease more strongly than epithelium without *H. pylori* CagA^+^, with lower values in preneoplasic lesions (AtL) (Figure 7). Although the AtL the expression of *GRN* fell in the presence and absence of *cagA*, the differences were not significant.

When we analyzed the gene expression levels of PGRN according to the presence of allelic profiles of *vacA* in the s1m1 allele, *GRN* was found to reach lower levels of gene expression than in the absence of this allele, presenting significant differences between both groups (*p* = 0.030), and in only the NL epithelium (*p* = 0.043) (Figure 8B). Regarding the *vacA* s2m2^+^ alleles, we did not observe important differences when comparing the levels of *GRN* in tissues colonized by strains that express, or do not express, this allele. When observing the tissue status of s2m2^+^ strains, the *GRN* remains equally expressed in erosive epithelia (EL) and non-erosive epithelia (NEL). Furthermore, the *GRN* expressions were even higher than those expressed in NL tissues, although they were non-significant. 

Finally, in the presence of *iceA^+^* (Figure 9), *GRN* expression was lower than observed by IceA^−^. In epithelia infected with strain carrying IceA^+^, the GRN expression presented a significant difference between NL and AtL epithelium (*p* = 0.021). In NEL and AtL, the expression of *GRN* was low, a trend previously described according to the other virulence genes analyzed. 

## 4. Discussion 

*H. pylori* infection is characterized by its chronicity [7]. In general, the infection is acquired in childhood. It may remain and asymptomatically colonize the gastric epithelium for decades or generate inflammatory manifestations in a small group of infected people, mainly in adults [27]. It is important to mention that the prevalence of lesions observed in our study group correlates with that expected for *H. pylori* infection. Of the infected dyspeptic patients (case), 61.33% (*n* = 46) did not present lesions observable by endoscopy, 18.67% presented EL and 14.67% presented NEL, while pre-neoplasia lesions reached a prevalence of 5.3% (*n* = 4), all in agreement with previous epidemiological reports [28,29,30]. 

The environment, characteristics of the population, and the virulence of the bacteria are critical factors for the development of the gastric disease [2]. Clinical and family history may impact the development of advanced pathologies, including cancer [31], and interestingly, the participants had an average age of 48.18 years. It is believed that *H. pylori* infection is usually acquired at an early age (before 10 years old), and is characterized by chronicity. In this case, a large proportion of this group would already be affected by a chronic process for more than three decades, a sufficient period for colonization and to generate changes in the gastric mucosa [5]. However, it is crucial to mention that the gastric epithelium is exposed to multiple factors that can mediate between health and disease. Transmissible factors, such as bacteria, protozoa, fungi and viruses, commensals, and pathogens, may induce inflammatory responses and cause epithelial damage [32]. Additionally, non-transmissible factors may affect these responses. Biliary or pancreatic secretions, diet, consumption of alcohol, drugs (e.g., non-steroidal anti-inflammatory drugs, NSAIDs), and some immunological disorders, may cause inflammation and could be reflected in gastric tissue alterations [33]. This is observable in tissues uninfected by *H. pylori*, such as our control group (controls).

Regarding the pathogenic process of *H. pylori*, the mechanisms that lead to the development of precancerous lesions and cancer are not fully understood [34]. However, diverse regulatory mechanisms, including progranulin, could occur to regulate the inflammatory process involved in lesion development [14,16]. Different functions are attributed to progranulin, depending on complete or cleaved status. In its complete form, the anti-inflammatory capacity of progranulin is mainly associated with the inhibition of TNFα through binding TNF receptors [35,36]. On the other hand, TNF-α is a proinflammatory cytokine that plays an important role in tumorigenesis by the gene-related expression of cytokines, adhesion molecules, and proangiogenic molecules [37]. Therefore, its presence indicates a poor prognosis for increased cancer development, including gastric cancer [38]. In the gene expression of PGRN in the antral tissue, no significant differences were observed, regardless of *H. pylori* infection or the presence of lesions; however, this behavior is different when evaluating specific types of gastric lesions. In the tissue colonized by *H. pylori* (case), the levels of *GRN* expression tend to decrease as the lesions progress by the Correa precancerous cascade [39]. In these infected tissues, the NL or EL expressed similar levels of *GRN*, which decreased in the tissue with micronodular gastropathy (NEL), with a significant drop in AtL where the glandular epithelial was lost (*p* < 0.05).

PGRN participates in the repair of damaged tissue in gastric ulcers. In fact, PGRN administered experimentally in fresh skin wounds of rats stimulates specific pathways of inflammation, and favors the accumulation of fibroblasts and vascular regeneration, which are necessary conditions for the repair of damaged tissue [21]. Studies of PGRN and the inflammatory response in gastric ulcers in murine models, attribute a role in the healing of gastric ulcers to PGRN by macrophages dependent on the macrophage colony-stimulating factor (M-CSF), promoting angiogenesis through the upregulation of Cyclooxygenase/Prostaglandin E (COX-2/PGE) production and expression of VEGF (Vascular Endothelial Growth Factor), [40]. Furthermore, studies of muscle injury regeneration suggest that PGRN is involved in the regulation of macrophage kinetics for muscle regeneration [41]. Analyses in human gastric tissue demonstrated the presence of PGRN both in the gastric epithelium, as well as in immune cells, except in lymphoid follicles [22]. In our study, EL reached levels of *GRN* expression similar to tissues without lesions (NL), which does not occur in NEL, where *GRN* levels begin to decrease.

A study by Wex et al. (2011) recognized higher levels of protein in PGRN, by immunohistochemistry, in the glandular tissue and on the basis of foveola of the stomach that present dense inflammatory infiltrates and its expression is weak or practically null between the gastric pits [22]. The transition of the epithelium in response to *H. pylori* infection in the progress of the Correa cascade may modify the receptor to TNFα, on which progranulin has an antagonistic effect [16] or may alter mitogenic pathways MEK1/2 and p38, as proposed by Wang 2011 [14]. Thus, epithelial and immune response changes would affect the expression of *GRN* in LNE-type lesions or deeper lesions such as AtL, which could explain *GRN* downregulation in cases where the precancerous cascade progresses, as observed in our results.

On the other hand, Wex previously described a lower expression of *GRN* in infected tissues when compared with uninfected tissues [22]. This downregulation of *GNR* could be attributed to translational regulation exerted by *H. pylori* infection, but further investigation is needed to elucidate our understanding. These changes in the cell cycle caused by *H. pylori* can push on transcriptional factors and mitogenic pathways [42,43], and therefore impact the *GRN* expression and the integrity of the gastric epithelium, although not all cases will progress to disease and malignancy [43].

Bacterial populations of *H. pylori* are extremely variable, and the effect of virulence factors can impact the cellular response [44]. VacA is a virulence factor which the pathogenic effect depends on genotypes conformed by alleles s and m, i.e., s1m1 and s1m2 alleles, which are recognized due to their ability to induce vacuolization of colonized cells, leading to a more accentuated cell damage [8]. In our study, the presence of *vacA* was observed in all the strains analyzed and at all levels of epithelial damage. Additionally, the expression levels of *GRN* in epithelium with and without the VacA s1m1 allele were statistically different (*p* < 0.05). Coincidentally, this allele (VacAs1m1^+^) predominates in atrophic epithelia that also show a significant drop in *GRN* expression. Furthermore, the *vacA* s1m1 alleles trigger damage, intracellular modifications, induce apoptosis, and inhibit the activation and proliferation of T lymphocytes, as well as the modulation of the inflammatory cytokine and immunosuppression [8] [14]. VacA, by a mechanism not yet elucidated, usurps the lysosomal and autophagy pathways that are favorable for the survival, colonization, and chronicity of the bacteria [18], and interrupts endolysosomal traffic, inducing autophagosomal survival [45].

Other virulence factors of H. *pylori* strains also influence the epithelium transformation process. Among them, *iceA* was present in 50.67% (*n* = 38), whereas *cagA* was present in 45.33% (*n* = 34). Coincidentally, IceA1 has been linked to ulcerative lesions when combined with CagA. CagA^+^IceA2^+^ is more dominant in active chronic inflammation, gastric ulcer, and carcinoma than when combined with the *A1* allele, according to studies in dyspeptic patients from Pakistan [46]. In our reports, *cagA* is present in atrophic lesions similarly to *vacA* s1m1 alleles (Appendix A). Studies have shown that infection by CagA^+^ strains is associated with cell cycle alterations, apoptosis, pronounced levels of inflammation, and greater intensity of gastric atrophy and metaplasia [47,48]. Regarding CagA and its relationship with epithelium status, Tserentogtokh et al. (2019) reported a prevalence of CagA of at least 30% in peptic ulcers, 41.6% in gastritis, and 64.7% in gastric cancer for western variant strains, and of up to 100% in isolates from Asian strains through sequencing [49]. In our study, CagA strains were isolated from all epithelia studied with predominance of the erosive type (*n* = 8, 57.14%).

CagA has been closely linked to malignant lesions, and is related to other virulence factors, with a tendency to migrate towards the healthy epithelium [50]. However, Wang et al., 2011, showed that bacterial virulence, and CagA overexpression did not induce *GRN* mRNA levels from the cDNA amplifier in cell lines immortalized by q-PCR, suggesting that CagA is not sufficient to induce PGRN upregulation [14]. In fact, according to our results, CagA does not influence PGRN gene expression.

CagA transcription factors increased cell proliferation through the Ras/MERK/ERK mitogenic pathway, as with PGRN [14,51]. Additionally, Cag A can induce the mitogenic response by the JAK2/STAT3 signaling pathway when it binds to the membrane protein GP130 and to SHP-2 [52,53]. Nevertheless, we did not find a link to PGRN. Other researchers also point out that CagA regulates inflammatory responses through the activation of *IL-8*, [54], however, no reports were found linking PGRN with the activation of this cytokine. This may explain the lack of association of CagA with the gene expression of PGRN. On the other hand, CagA induces inflammation through the activation of NF-κB by way of tumor necrosis factor alpha (TNF-α) [55,56], a pathway also controlled by PGRN [16]. This common utilization of inflammatory pathways is possibly due to a competition effect generated between TNF-α and PGRN molecules. Further studies are needed to corroborate this interaction in future studies, considering that the samples analyzed using these epithelial characteristics are scarce compared to the status of other epithelial lesions.

Additionally, IceA^+^ is also found in atrophic lesions, presenting significant differences between the levels of *GRN* detected in the epithelium without lesions and the atrophic one (*p* < 0.05). However, the pathogenesis of molecular pathways of IceA still need to be elucidated.

Finally, downward deregulation in *GRN* levels is observed in the most severe lesions, coincidentally colonized by more aggressive strains of *H. pylori*. Although the data are interesting, it is essential to consider the limitations of our results, particularly the bias from the lesion identification criteria and the few participants with atrophic and advanced lesions. These factors contributed to a wide dispersion of the data, making it necessary to increase the sample size. Further studies are needed to focus on the regulatory mechanisms of PGRN and the *GRN* expression as a factor in the inflammatory process and cellular response associated with the infectious process.

## 5. Conclusions

Although the levels of *GRN* expression detected in *H. pylori*-infected tissues are similar to those obtained in uninfected tissue, the *GRN* expression progressively decreases according to lesion progression in the precancerous Correa’s cascade, mainly in the tissues colonized by *H. pylori* with a lower *GRN* expression in the more advanced stages of the lesion. With *H. pylori* infection, *GRN* expression is affected by the virulence of the infecting strains, particularly *vacA* s1m1 *and IceA1*. However, *cagA* did not show significant effects. Additional research should consider molecular processes occurring at the immune response level and regulatory pathways of the cell cycle associated with VacA and CagA strains.

## Figures and Tables

**Figure 1 microorganisms-10-00998-f001:**
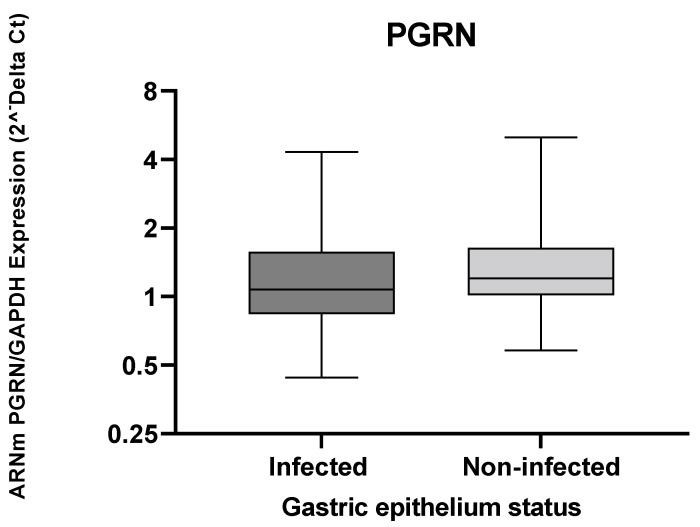
The *GRN* expression in gastric tissue from individuals infected and uninfected with *H. pylori*. The *GRN* expression values calculated by formula 2^−ΔCt^, where ΔCt = Ct of *GRN* − Ct of *GAPDH.* The Mann–Whitney test was used for the analyses between the groups. Statistical significance *p* < 0.05.

**Figure 2 microorganisms-10-00998-f002:**
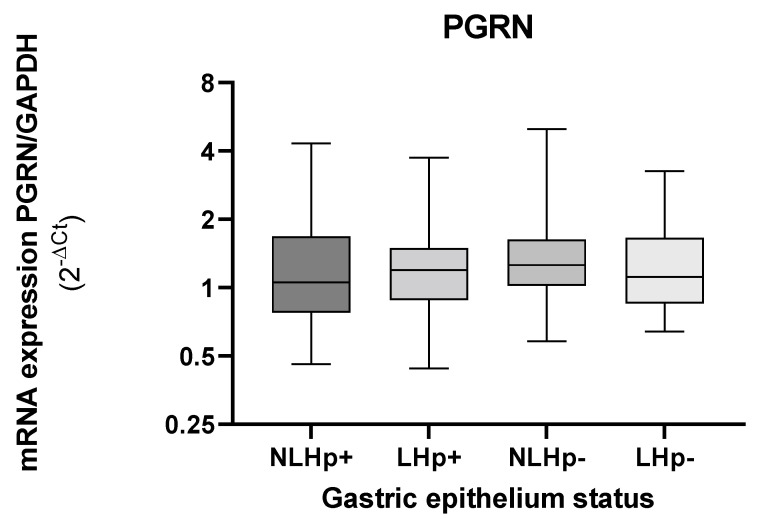
Expression levels of *GRN* mRNA in gastric biopsies according to tissue status infected (Hp+) and non- infected with *H. pylori* (Hp−). Tissues with injury (L), tissues without injury (NL). The *GRN* expression values calculated by formula 2^−ΔCt^, where ΔCt = Ct of *GRN* − Ct of GAPDH. The status of the gastric tissue was assigned according to the report of the endoscopic observation. The Kruskal–Wallis and Mann–Whitney was used for the analysis in or between the groups. Statistical significance < 0.05.

**Figure 3 microorganisms-10-00998-f003:**
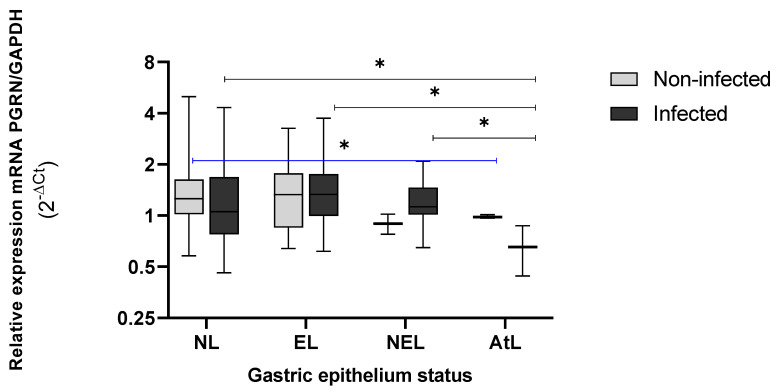
Expression levels of *GRN* mRNA in gastric tissue with and without lesions from individuals infected (*n* = 75) and non-infected with *H. pylori* (*n* = 75). NL = non lesions, EL = erosive lesion, NEL= non-erosive lesion, AtL= atrophic lesion. The PGRN expression values were calculated by formula 2^−ΔCt^, where ΔCt = Ct of *GRN* − Ct of *GAPDH*. The status of the gastric tissue was assigned according to the endoscopic observation report. The Kruskal–Wallis and Mann–Whitney tests were used for analysis in or between the groups. * Statistical significance was established at < 0.05.

**Figure 4 microorganisms-10-00998-f004:**
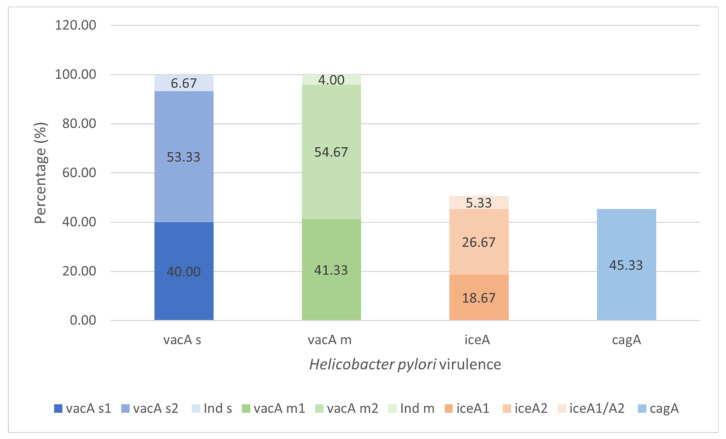
Prevalence of the virulence VacA, CagA, and IceA of the colonizing *H. pylori* strains in gastric tissue (*n* = 75). Bacterial DNA amplification for the *vacA*, *cagA* and *iceA* by the PCR (Polymerase Chain Reaction) test and PCR-RFLP (Restriction Fragments Length Polymorphism) assay to confirm the *s* and *m* alleles of *vacA*, visualized in gel of agarose 1.5% (Cleaver Scientific, Rugby, UK). Ind *s* or Ind *m*, refer to *alleles* s or m of Vac non-specified by PCR or PCR-RFLP methods. Values represent a percentage of the totals.

**Figure 5 microorganisms-10-00998-f005:**
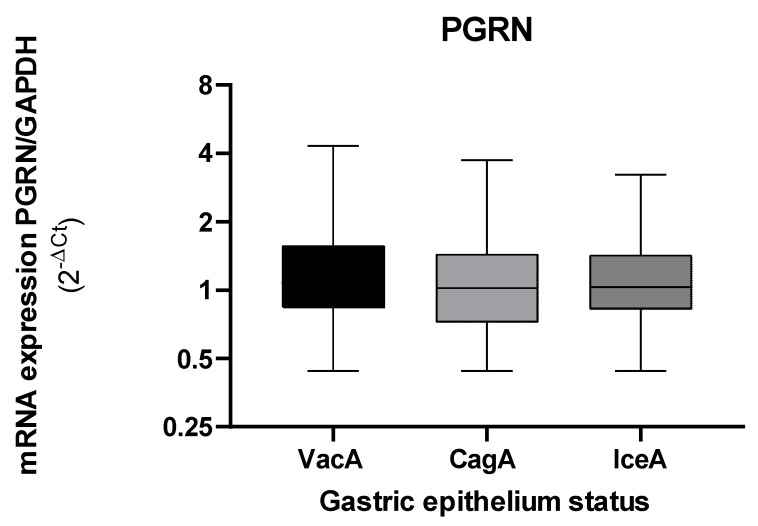
Expression levels of *GRN* mRNA in gastric tissue according to the virulence of *H. pylori* bacteria that present the *vacA, cagA,* and *iceA*. The *GRN* expression values calculated with the following formula: 2^−ΔCt^, where ΔCt = Ct of *GRN* − Ct of *GAPDH.* Virulence factors were calculated by an amplification of the *vacA*, *cagA*, and *iceA* genes with the PCR method. The Kruskal–Wallis and Mann–Whitney tests were used for the analysis in or between the groups. Statistical significance was established at <0.05.

**Figure 6 microorganisms-10-00998-f006:**
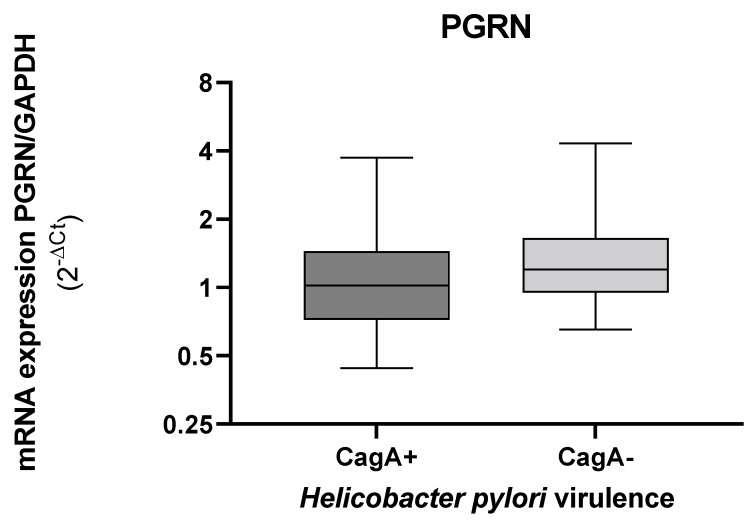
Expression levels of *GRN* mRNA in gastric tissue according to the presence or absence of the *H. pylori cagA*. The *GRN* expression values calculated by the formula 2^ΔCt,^ where ΔCt = Ct of *GRN* − Ct of *GAPDH*. Bacterial DNA amplification for the *cagA* and by the PCR (Polymerase Chain Reaction), visualized in gel of agarose 1.5% (Cleaver Scientific. Rugby, UK). Data compared by non-parametric Mann–Whitney test. Statistical significance *p* < 0.05.

**Figure 7 microorganisms-10-00998-f007:**
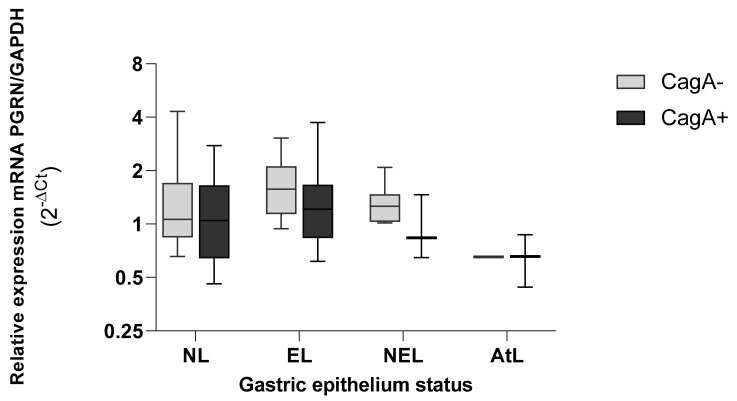
Expression levels of *GRN* mRNA in gastric tissue according to the virulence load of *H. pylori* CagA+ bacteria. NL = non lesions, EL = erosive lesion, NEL = non-erosive lesion, AtL = atrophic lesion. PGRN expression values calculated by the formula 2^−ΔCt^, where ΔCt = Ct of *GRN* − Ct of *GAPDH*. The virulence of *H. pylori* was detected by amplifying the *cagA* gene with the PCR method. The status of the gastric tissue was determined according to the endoscopic observation report. Data compared by non-parametric Kruskal–Wallis test and comparison between two variables by Mann–Whitney. Statistical significance was established at *p* < 0.05.

**Figure 8 microorganisms-10-00998-f008:**
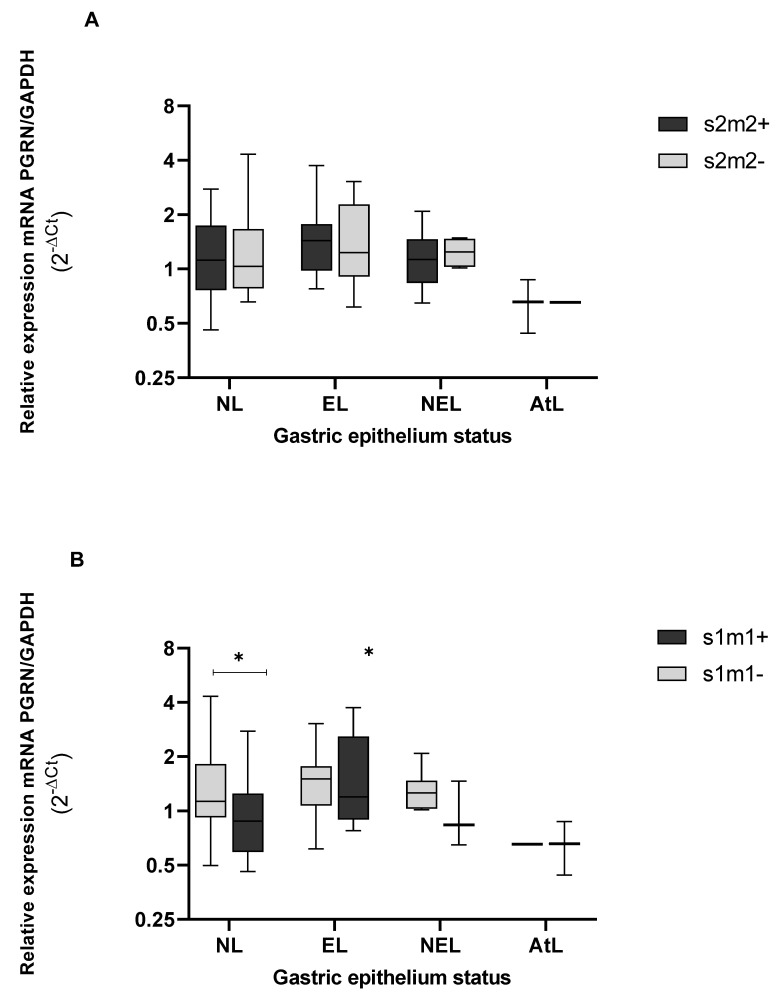
Genic expression of PGRN in the different status of gastric tissue colonized with *H. pylori* vacAs2m2^+^, (**A**) and vacA s1m1^+^. (**B**) NL = epithelium without lesions; EL = erosive lesions; NEL = non-erosive lesions; AtL = atrophic lesions. Calculated *GRN* expression values by the formula 2^−ΔCt^, being ΔCt = Ct of *GRN* − Ct of *GAPDH*. The virulence of *H. pylori* was detected by amplification of the *vacA* gene, s and m alleles, by the PCR method, and the PCR polymorphic restriction assay-RFLP for confirmation of vacA s and m alleles, visualized on agarose gel (Cleaver Scientific, Rugby, UK). The status of the gastric tissue was determined according to the report of the endoscopic observation. The data were compared by a non-parametric Kruskal–Wallis and Mann–Whitney tests for comparison between two variables. * Statistical significance *p* < 0.05.

**Figure 9 microorganisms-10-00998-f009:**
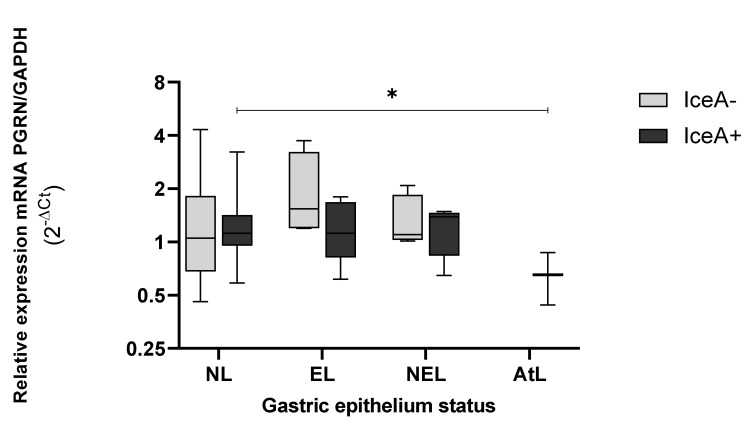
Genic expression of *GRN* in the different status of gastric tissue colonized with *H. pylori* iceA^+^. NL = epithelium without lesions; EL = erosive lesions; NEL = non-erosive lesions; AtL = atrophic lesions. Calculated *GRN* expression values by the formula 2^−ΔCt^, being ΔCt = Ct of *GRN* − Ct of *GAPDH*. The virulence of *H. pylori* was detected by amplification of the *iceA* gene, by the PCR method, visualized on agarose gel (Cleaver Scientific, Rugby, UK). The status of the gastric tissue was determined according to the endoscopic observation report. The data was compared by non-parametric Kruskal–Wallis and Mann–Whitney tests for comparison between two variables. * Statistical significance *p* < 0.05.

**Table 1 microorganisms-10-00998-t001:** Sociodemographic characteristics of the participants infected and non-infected with *H. pylori* (*n* = 150).

	Total, *n* = 150	Infected (Cases) *n* = 75	Non Infected (Controls) *n* = 75	
Background	*n* (%)	*n* (%)	*n* (%)	*p* Value
Age (years ± SD)	48.18 ± 14.26	48.23 ± 13.19	48.13 ± 15.35	0.968
Males	48 (32.00)	24 (32.00)	24 (32.00)	0.999
Females	102 (68.00)	51 (68.00)	51 (68.00)	0.999
Ethnicity Mapuche	46 (30.67)	23 (30.67)	23 (30.67)	0.999
Rurality	32 (21.33)	16 (21.33)	16 (21.33)	0.999
Health centers Public	95 (63.33)	52 (69.33)	43 (57.33)	0.175
Education level < 12 years	86 (57.33)	44 (58.67)	42 (56.00)	0.869
Health Insurance National Insurance FONASA	117 (78.00)	59 (78.67)	58 (77.33)	0.999
Household (≥5 members)	21 (14.00)	12 (16.00)	09 (12.00)	0.640
Addictive habits				
-Smoker	43 (28.67)	20 (26.67)	23 (30.66)	0.590
-Drink Alcohol	72 (48.00)	37 (49.33)	35 (46.67)	0.870
-Family history of gastric cancer	47 (31.33)	21 (28.00)	26 (34.67)	0.482
Comorbidity				
-Diabetes	27 (18.00)	11 (14.67)	16 (21.33)	0.396
-Arterial hypertension	35 (23.33)	16 (21.33)	19 (25.33)	0.700
-Hypercholesterolemia	36 (24.00)	12 (16.00)	24 (32.00)	0.035 *
-Cardiovascular diseases	8 (05.33)	2 (02.67)	6 (08.00)	0.276
Others **	71 (47.33)	45 (60.00)	31 (41.33)	0.033 *
Without comorbidities	53 (35.33)	33 (44.00)	21 (28.00)	0.060 *

FONASA, from Spanish—“FOndo NAcional de SAlud” (public). *n* = number of individuals. (%) percentage of totals according to each group. Gastric tissue status according to endoscopic observation report. Values expressed as a percentage of the means, compared by Chi-square test. * Statistical significance, *p*-value < 0.05. ** Other diseases, including thyroid disorders, autoimmune diseases, and hepatic alterations.

**Table 2 microorganisms-10-00998-t002:** Gastric epithelium status of the infected (case) and non-infected with *H. pylori* (control) groups.

Epithelium	Total *n* = 150 *n* (%)	Infected *n* = 75 *n* (%)	Non-Infected *n* = 75 *n* (%)	*p* Value	Odds Ratio (CI)
Non-lesions (NL)	94 (62.67)	46 (61.33)	48 (64.00)	0.866	1.121 (0.474–1.773)
Lesions (L)	56 (37.33)	29 (38.67)	27 (36.00)	0.866	1.121 (0.564–2.112)
Erosive (EL)	35 (23.33)	14 (18.67)	21 (28.00)	0.029 *	0.267 (0.084–0.828)
Non-erosive (NEL)	13 (08.67)	11 (14.67)	2 (02.67)	0.010 *	7.640 (1.670–36.620)
Atrophic (AtL)	6 (04.00)	3 (04.00)	3 (04.00)	>0.999	0.923 (0.201–4.261)
Advanced (AdL)	2 (01.33)	1 (01.33)	1 (01.33)	>0.999	0.929 (0.047–18.26)

Reports of epithelium status were derived from endoscopic observation. * Statistical significance established at *p* < 0.05.

## Data Availability

Not applicable.

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
