# Peer review of "Association of Progranulin Gene Expression from Dyspeptic Patients with Virulent Helicobacter pylori Strains; In Vivo Model"

_microorganisms, 2022, doi:10.3390/microorganisms10050998_

Round 1
Reviewer 1 Report
Troncoso et al. presented a study study the gastric disease course caused by H. pylori infection, and their relationship between progranulin genic expression and virulence of H. pylori. The manuscript provides interesting insights into the topic and it could be considerer as a good initial point for future studies. The study is well thought out and minor changes must be done:
abstract: PGRN is abbreviated
Introducction: Currently, PGRN is considered a growth factor with higher.¿?¿?
In figure 2: Tissues with injury (CL), tissues without injury (SL) Which one is each?
The text is sometimes difficult to follow. Authors put full stops where they don't belong and use commas when a full stop is necessary. Please revise it
Author Response
Manuscript microorganisms-1655161
Title: Association of progranulin gene expression from dyspeptic patients with virulent Helicobacter pylori strains. In vivo model
Microorganisms
RESPONSE TO REVIEWERS’ COMMENTS:
REVIEWER #1
Overall Comments
English language and style
(x) English language and style are fine/minor spell check required
Are the results clearly presented?
(x) Can be Improved
Comments and Suggestions for Authors
Troncoso et al. presented a study study the gastric disease course caused by H. pylori infection, and their relationship between progranulin genic expression and virulence of H. pylori. The manuscript provides interesting insights into the topic and it could be considerer as a good initial point for future studies. The study is well thought out and minor changes must be done:
R: We would like to thank the reviewer for his/her comments and for all the suggestions for improving the manuscript, which were very useful. As suggested, the text was modified according to the reviewer’s suggestions, which are answered below in this letter. Moreover, the new version of the manuscript was reviewed by a native English speaker.
Query 1:
Abstract: PGRN is abbreviated
R: Thank you very much for your observation. The abbreviation of Progranulin in the abstract was replaced by the fully name of the protein and was used just after mentioned in the main text (Page 1).
Query 2:
Introduction: Currently, PGRN is considered a growth factor with higher???
R: The sentences was reformulated and reviewed by an English native speaker (Page 2). The new sentences was: ……“Currently, PGRN is considered a growth factor with high expression levels in hepatocellular, ovarian, bladder cancer, and glioblastoma, and is associated with a bad prognosis”…….
Query 3
In figure 2: Tissues with injury (CL), tissues without injury (SL) Which one is each?
R: We apologize for the mistake. The observation was considerate and modified by: Tissues with injury (L) and tissues without injury (NL) (Page 8).
Query 4
The text is sometimes difficult to follow. Authors put full stops where they don't belong and use commas when a full stop is necessary. Please revise it.
R: We thank the reviewer for this observation. As suggested all the manuscript was extensively reviewed by software of Grammar correction and for an English native speaker translator.
Reviewer 2 Report
“Association of progranulin gene expression from dyspeptic patients with virulent Helicobacter pylori strains. In vivo model” is an interesting article. However, I think some ameliorations are required.
- Language and punctuation: The paper needs an extensive editing of English language. In some paragraphs, reading and understanding are quite difficult.
- Abstract: Please specify PGNR acronym at its first mention.
- Table 1: Please add age and sex information of both subgroups (infected and non-infected). These data are essential to check the homogeneity of the two samples.
- Discussion: Please add a brief explanation of why non-infected patients presents epithelial lesions too.
- References: Please modify bibliography and citations according to editorial instructions.
Author Response
Manuscript microorganisms-1655161
Title: Association of progranulin gene expression from dyspeptic patients with virulent Helicobacter pylori strains. In vivo model
Microorganisms
RESPONSE TO REVIEWERS’ COMMENTS:
REVIEWER #2
Overall Comments
English language and style
(x) Extensive editing of English language and style required
Does the introduction provide sufficient background and include all relevant references?
(x) Can be improved
Is the research design appropriate?
(x) Can be improved
Are the results clearly presented?
(x) Can be improved
Are the conclusions supported by the results?
(x) Can be improved
Comments and Suggestions for Authors
Association of progranulin gene expression from dyspeptic patients with virulent Helicobacter pylori strains. In vivo model” is an interesting article. However, I think some ameliorations are required.
R: We would like to thank the reviewer for his/her comments and for all the suggestions for improving the manuscript, which were very useful. As suggested, the text was modified according to the reviewer’s suggestions, which are answered below in this letter. Moreover, the manuscript was extensively reviewed by a native English speaker and modified for a better and more clarifying reading.
Query 1:
Language and punctuation: The paper needs an extensive editing of English language. In some paragraphs, reading and understanding are quite difficult.
R: We thank the reviewer for this observation. As previously mentioned, all the manuscript was extensively reviewed by software of Grammar correction and for an English native speaker translator.
Query 2:
Abstract: Please specify PGNR acronym at its first mention.
R: The abbreviation of Progranulin in the abstract was replaced by the fully name of the protein and was used just after mentioned in the main text. (Page 1)
Query 3
Table 1: Please add age and sex information of both subgroups (infected and non-infected). These data are essential to check the homogeneity of the two samples.
R: As suggested, in the table 1 was included the age and sex information of each group. (Page 5)
Query 4
Discussion: Please add a brief explanation of why non-infected patients presents epithelial lesions too.
R: An explanation of why non-infected patients presents epithelial lesions was added in the discussion (Page 14). The explanation was the following: “…..However, it is crucial to mention that the gastric epithelium is exposed to multiple factors that can mediate between health and disease. Transmissible factors, such as bacteria, protozoa, fungi and viruses, commensals, and pathogens, may induce inflammatory responses and cause epithelial damage. Also, the non-transmissible factors may affect these responses. Biliary or pancreatic secretions, diet, consumption of alcohol, drugs (e.g., non-steroidal anti-inflammatory drugs, NSAIDs), and some immunological disorders, may cause inflammation and could be reflected in gastric tissue alterations. This is observable in tissues not infected by H. pylori, such as our control group (controls)….”
Query 5
References: Please modify bibliography and citations according to editorial instructions.
R: We thank the reviewer for this observation. As suggested, all the references were reviewed and modified according to editorial instruction.